# Extracellular modulation of TREK-2 activity with nanobodies provides insight into the mechanisms of K2P channel regulation

Karin E. J. Rödström[1,2,3,4,11], Alexander Cloake[1,11], Janina Sörmann[1,2,11], Agnese Baronina[3,11], Kathryn H. M. Smith[1,2,4], Ashley C. W. Pike[3], Jackie Ang[3], Peter Proks[1,2], Marcus Schewe[5], Ingelise Holland-Kaye[4], Simon R. Bushell[3], Jenna Elliott[1], Els Pardon[6,7], Thomas Baukrowitz[5], Raymond J. Owens[8,9], Simon Newstead[2,4,10], Jan Steyaert[6,7], Elisabeth P. Carpenter[3,10] ✉ & Stephen J. Tucker[1,2,10] ✉

Potassium channels of the Two-Pore Domain (K2P) subfamily, *KCNK1-KCNK18*, play crucial roles in controlling the electrical activity of many different cell types and represent attractive therapeutic targets. However, the identification of highly selective small molecule drugs against these channels has been challenging due to the high degree of structural and functional conservation that exists not only between K2P channels, but across the whole K⁺ channel superfamily. To address the issue of selectivity, here we generate camelid antibody fragments (nanobodies) against the TREK-2 (*KCNK10*) K2P K⁺ channel and identify selective binders including several that directly modulate channel activity. X-ray crystallography and CryoEM data of these nanobodies in complex with TREK-2 also reveal insights into their mechanisms of activation and inhibition via binding to the extracellular loops and Cap domain, as well as their suitability for immunodetection. These structures facilitate design of a biparatropic inhibitory nanobody with markedly improved sensitivity. Together, these results provide important insights into TREK channel gating and provide an alternative, more selective approach to modulation of K2P channel activity via their extracellular domains.

Ion channels control or contribute to the electrical activity and function of almost every living cell. Their selective pharmacological manipulation therefore offers wide-ranging approaches to plant, animal and human health[1]. In particular, the regulation of K⁺-selective ion channels presents therapeutic potential for a wide variety of human disorders including epilepsy, arrhythmias and various other cardiovascular diseases and neurological conditions, especially the control of pain. Significant effort has therefore been expended in the identification of drugs that target specific types of K⁺ channels for the treatment of such disorders[2,3].

However, although there are over 80 different K⁺ channel genes in the human genome, most have a similar architecture in their pore

[1]Clarendon Laboratory, Department of Physics, University of Oxford, Oxford, UK. [2]Kavli Institute for Nanoscience Discovery, University of Oxford, Oxford, UK. [3]Centre for Medicines Discovery, University of Oxford, Oxford, UK. [4]Department of Biochemistry, University of Oxford, Oxford, UK. [5]Institute of Physiology, Medical Faculty, Kiel University, Kiel, Germany. [6]Structural Biology Brussels, Vrije Universiteit Brussel, Brussels, Belgium. [7]VIB-VUB Center for Structural Biology, VIB, Brussels, Belgium. [8]The Rosalind Franklin Institute, Harwell Campus, Didcot, UK. [9]Division of Structural Biology, University of Oxford, Oxford, UK. [10]OXION Initiative in Ion Channels and Disease, University of Oxford, Oxford, UK. [11]These authors contributed equally: Karin E. J. Rödström, Alexander Cloake, Janina Sörmann, Agnese Baronina. ✉e-mail: lizcarpen1@gmail.com; stephen.tucker@physics.ox.ac.uk

region because this highly conserved structural feature facilitates the selective permeation of K[+] as well as some of the conserved gating mechanisms they employ. As a consequence, the identification of highly selective small molecule agonists and/or antagonists that exhibit no cross-reactivity with other K[+] channels is challenging[1,4].

Two-Pore Domain K[+] channels (K2P) are a distinct subset of K[+] channels where each gene encodes a subunit with two pore-forming domains that dimerise to create a single pseudo tetrameric K[+] selective channel across the membrane[5]. These channels underlie the background K[+] currents that control the membrane potential in many different cell types where their activity is often regulated by diverse stimuli[6]. In particular, the TREK subfamily of K2P channels (TREK-1, TREK-2 and TRAAK) are regulated by a wide variety of physical and chemical stimuli. This polymodal regulation allows them to integrate cellular electrical activity with a diverse range of cellular signalling pathways[2,7].

A number of structures now exist for members of the TREK subfamily that has allowed detailed gating models to be proposed[4,8–11]. Like many other K[+] channels, movement of the pore-lining helices, in particular the M4 helix appears important for the regulation of TREK channel gating[9,12]. However, unlike many other K[+] channels, these movements do not constrict a lower 'bundle-crossing-like' gate at the cytoplasmic entrance to the pore[9]. Instead, the channel appears to be gated primarily by much smaller structural changes in the selectivity filter with the 'M4 Up' conformation supporting a more active state of the channel than the M4 down conformations. However, there is evidence that the channel can support K[+] conduction in both conformations[13,14]. Recent evidence suggests that the movement of M4 couples to the filter gate via changes in the network of interactions that support the selectivity filter in its activated state[15].

The expression of TREK channels within the central and peripheral nervous systems[7] and their involvement in processes such as pain perception, neuroprotection, and anaesthesia makes them attractive therapeutic targets[2,16]. Recent approaches have focused on the development of TREK channel agonists as potential analgesics and several lead compounds have been identified that target different elements of the gating mechanism within the selectivity filter of the channel[4,11,17]. However, although such small molecules have proven to be highly effective agonists, and much is now known about their mechanism of action, they often exhibit cross-reactivity within the TREK subfamily. Also, in many cases, these agonists activate other classes of K[+] channel thereby limiting their potential use[4]. Other approaches to target these channels may therefore be required.

For some classes of ion channels, the adaptation of channel toxins and the development of RNA-based antagonists has been successful in achieving greater target selectivity[18,19], but perhaps the most promising approaches involve biologics that exploit protein-protein interactions for target recognition[20–23]. Many antibody-based therapies are now in clinical use for the treatment of a variety of disorders and have a number of potential benefits compared to traditional approaches although their identification and development can be costly[24].

Nanobodies are single-domain (V$_{HH}$) antibodies derived from camelid heavy chain antibodies[25]. Their name derives from their small size and when compared to small molecule inhibitors, have certain advantages such as high target specificity and low toxicity leading to fewer off-target effects[26]. In addition, their small size and long complementarity determining region 3 (CDR3) loops enable them to access cavities or clefts in proteins that are inaccessible to antibodies[27].

They can also exhibit greater stability and solubility than conventional antibodies, so when combined with their relative ease of production, purification and engineering, this makes them ideal candidates for many therapeutic strategies where small molecule approaches have stalled[28], as well as equivalent detection surrogates for antibodies in immunocytochemistry[29]. They are also particularly

useful probes to study structure-function relationships and other properties of ion channels and membrane proteins in general[30].

In this study, we generate a range of nanobody binders to the TREK-2 K2P channel, several of which exhibited highly specific regulation of channel activity. Our results also permit the design of a biparatropic inhibitory nanobody targeting spatially discrete epitopes which exhibited markedly improved sensitivity. The insights provided by these nanobodies highlight mechanistic approaches to the study of K2P channels and their role in vivo, and expand the potential of nanobodies as therapeutic agents.

## Results and discussion
### Generation of TREK-2 nanobodies
Nanobodies were produced as part of the Nanobodies4Instruct program in a method described by Pardon et al. [31]. Briefly, purified human TREK-2 protein, identical to that used in previous structural studies[9] was used to elicit an immune response in llamas. The resulting immune repertoire was isolated and positive clones were selected by phage panning and screening against the same TREK-2 protein. Overall, a panel of 29 unique nanobodies were produced (Supplementary Fig. S1) and these were further examined for their ability to produce a peak shift on size exclusion chromatography (SEC) to determine which nanobodies were capable of forming tight complexes (Fig. 1A). These tight binders were considered more suitable for subsequent structural and/or functional applications.

### Screening for functional effects
To determine the possible functional effects of the nanobodies on channel activity, human TREK-2 channels were expressed in *Xenopus* oocytes and currents measured before and after the extracellular addition of nanobody (0.5 µM). An increase (or decrease) in the current size by more than 50% was used as the initial threshold in this screen. The results are shown in Fig. 1A and reveal that from this initial panel, three exhibited clear functional modulation of TREK-2: nanobody CA10761 inhibited channel activity, whilst CA10767 and CA10776 both activated TREK-2. Functional effects of these nanobodies were also consistent with their tight binding as determined in the SEC-shift assay (Fig. 1A). These modulatory nanobodies were then selected for further structural studies. Another tight binder that exhibited no functional effect (CA10758) was also included.

### Nanobody binding to the extracellular Cap domain
Structures of these four nanobodies in complex with TREK-2 were then determined by X-ray crystallography using approaches similar to those previously described[4,9] and were resolved between 2.4 Å and 3.6 Å resolution. For clarity, these nanobodies were also renamed to reflect their functional effect (Fig. 1B and Supplementary Fig. S2 and Table S1).

All four of these nanobodies exhibited binding to the extracellular Cap domain of TREK-2 (Fig. 1B). Interestingly, the functionally inactive binder only interacted with the extracellular tip of the Cap domain whilst the modulatory nanobodies interact with both the Cap and extracellular loops much closer to the selectivity filter where the primary gate is located. In one case (CA10767), binding was also observed to the intracellular face as well as the Cap domain. The properties of these structures and the functional effects of each modulatory nanobody are described in more detail below.

### Immunodetection of TREK-2 expression at the cell surface
Previous studies have demonstrated the usefulness of nanobodies for immunodetection as well as in various immunoassays and other nanobody-based protein targeting strategies[20,29]. The structures determined above show that these TREK-2 nanobodies bind to the extracellular Cap domain indicating that immunodetection of TREK-2 at the cell surface should be possible. To examine this, we chose the

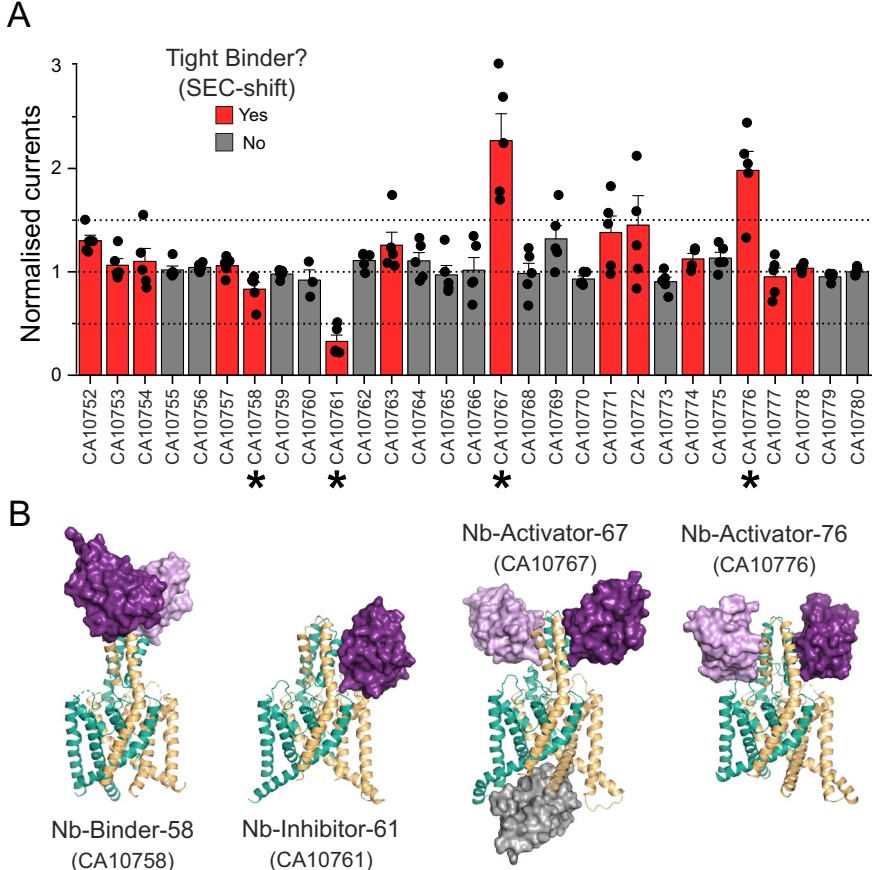

**Fig. 1 | Identification of functionally active nanobody binders for TREK-2.**
**A** Whole-cell currents recorded (at +40 mV) from oocytes expressing TREK-2 in the presence of a panel of nanobodies generated against TREK-2. Currents in the presence of extracellular nanobody (0.5 μM) were normalised to those in its absence. A threshold of 50% activation or inhibition was set in this screen (dotted lines). Tight binders within this panel were identified on the basis of producing a shift in size exclusion chromatography (SEC-shift) and shown in red. Asterisks mark the functionally active nanobodies plus an inactive tight binder chosen for further investigation. (Error bars represent mean ± S.D; $n \geq 5$) **B** Crystal structures of the three functionally active nanobodies in complex with TREK-2. The nanobodies are shown in purple as surface representations whilst TREK-2 is shown in cartoon (green and gold). For CA10767 (Nb-Activator-67) the third 'intracellular' nanobody in the unit cell is shown in grey. A structure of the functionally inactive binder, CA10758 (Nb-Binder-58) is also shown in complex with TREK-2. In this case the nanobody binds as a dimer to the top of the Cap domain away from the extracellular TM/Pore loops that influence channel gating.

functionally inactive nanobody CA10758 which interacts with the most exposed tip of the Cap domain. This nanobody is hereafter referred to as Nb-Binder-58 (or Nb58).

Many K2P channels exhibit N-linked glycosylation to their extracellular surfaces that may affect nanobody binding[32]. However, only one glycosylation site exists within TREK-2 in the extracellular loop that precedes Pore-Helix-1 (PH1). None of the four chosen nanobodies interact with or near this loop. Instead, Nb-Binder-58 binds as a dimer to the very tip of the Cap domain where it interacts with several residues that are unique between TREK-2 and the closely related TREK-1 channel (Fig. 2A).

To determine whether this nanobody can detect TREK-2 expression at the cell surface, Nb-Binder-58 was fused to the mVenus fluorescent protein and applied to HEK-293 cells expressing TREK-2 tagged with the mCherry fluorescent protein. Figure 2B shows overlapping signals that are not observed on cells expressing TREK-1, nor for any other K2P channel (Supplementary Fig. S3). A similar, functionally inactive nanobody (CA10757: Nb-Binder-57 or Nb57) that exhibits almost complete sequence identity with Nb-Binder-58 is also capable of similar immunodetection of TREK-2 (Fig. 2B and Supplementary Figs. S1 and S3). This indicates that both these nanobodies represent useful tools for the specific immunodetection of homomeric TREK-2 channels.

## A partially-selective activatory nanobody

The first activatory nanobody (CA10767) hereafter referred to as Nb-Activator-67 (or Nb67), binds to the side of the Cap domain (Fig. 3A) with all three CDR loops interacting with the external face of the Cap helices. However, these interactions are not symmetrical and on one side the EH1 Cap helix is partially unwound at the midpoint (residues 107–109; Fig. 3C). The structure also reveals a nanobody bound to the intracellular side where it occludes the entrance to the inner cavity (Fig. 3A). However, upon closer examination of the crystal structure this 'intracellular' interaction appears to result from tetramerisation of the nanobody within the unit cell that may drive the crystal packing and the resulting non-canonical structure (Supplementary Fig. S4A). Nevertheless, to rule out any functional effects of this intracellular interaction we examined the effect of this nanobody on TREK-2 currents in excised patches where it could be applied to the intracellular surface of the membrane. No functional effect on TREK-2 activity was observed (Supplementary Fig. S4B). This is in marked contrast to its clear activatory effect when applied extracellularly (Fig. 1A). Further examination revealed that Nb-Activator-67 rapidly increased channel activity ~3–4 fold with nanomolar efficacy (EC$_{50}$ = 101 ± 12 nM) when perfused onto oocytes expressing human TREK-2 (Fig. 3B).

To address its selectivity, we examined effects on other K2P channels. No functional modulation was observed on either TALK-2 or

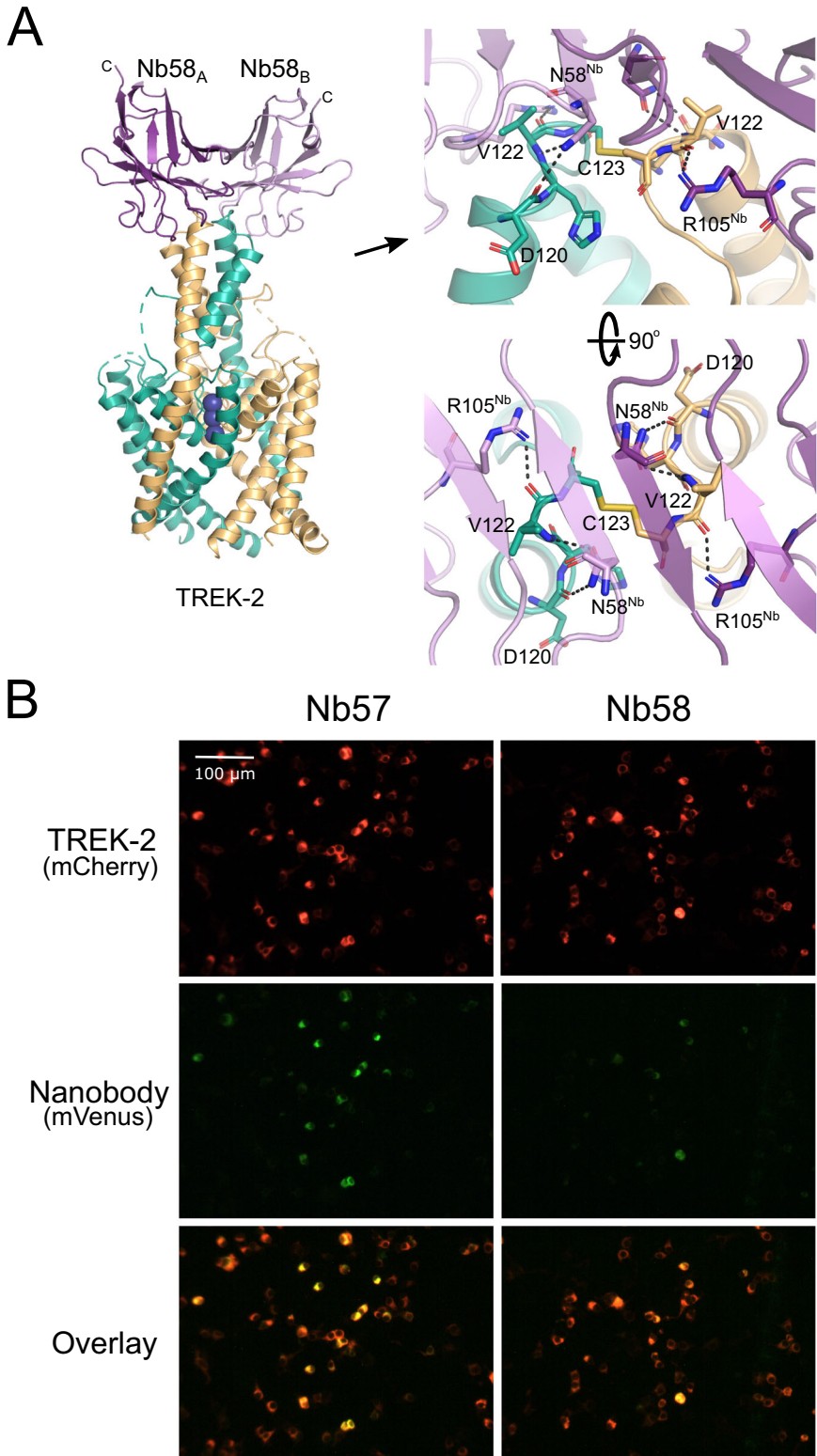

**Fig. 2 | Nb-binder-58 interacts with the apex of the Cap domain. A** Crystal structure of Nb-binder-58 (Nb58) in complex with TREK-2. This functionally inactive tight binder interacts as a dimer with the tip of the Cap domain only. The relative position of the C terminus of each Nb58 chain is indicated. The right-hand panel shows an expanded view of the interactions involved and shows the C123 disulphide between chains at the apex of the Cap. D120 and V122 are unique in TREK-2 compared to other TREK channels. **B** Immunodetection of TREK-2 at the cell surface. TREK-2-mCherry fusions were expressed in HEK-293 cells and stained with either Nb58 (or the highly related Nb57) fused to mVenus via the C-terminus indicated in panel A. Overlay of fluorescent signals shows clear binding at the extracellular surface. No overlapping signals are seen in untransfected controls or with other K2Ps (see Supplementary Fig. S3). These results were repeatable in $n \geq 3$ independent replicates.

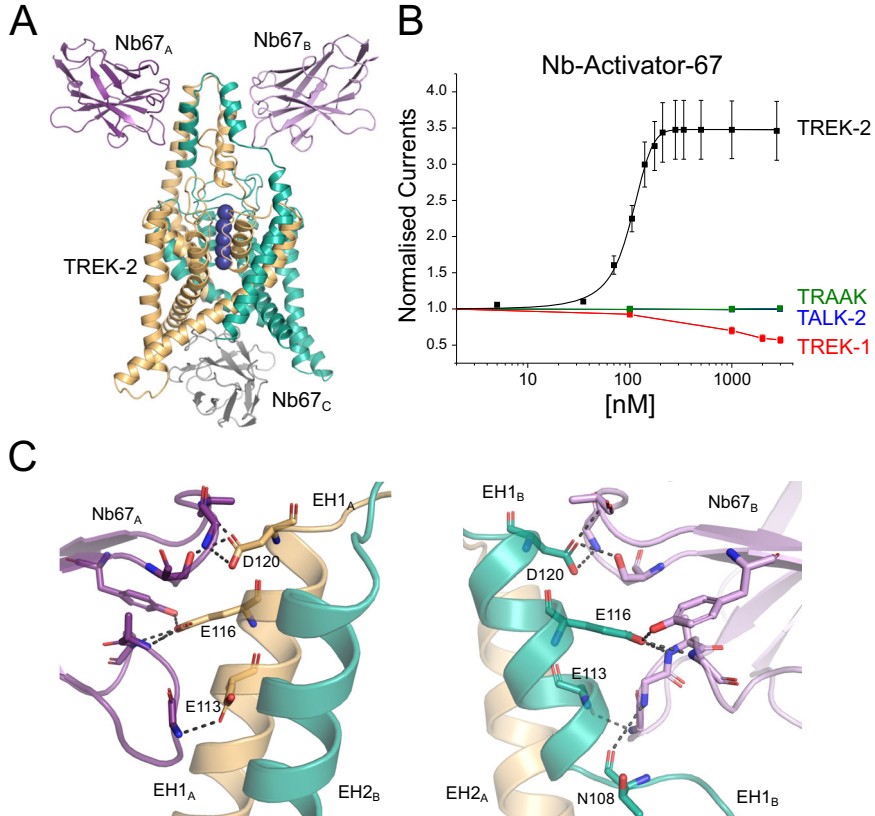

**Fig. 3 | Extracellular activation of TREK-2 by Nb-Activator-67. A** Crystal structure of TREK-2 in complex with Nb-Activator-67 (Nb67). Nanobodies bind to each side of the Cap domain and on one side (green subunit) the Cap helix unwinds. A third Nb67 (grey) is also found in the asymmetric unit bound to the intracellular surface. These unusual features may arise from the tetramerisation of Nb67 within the crystal (see Supplementary Fig. 4A). **B** Dose-response relationships showing extracellular activation of TREK-2 channel activity by Nb67. Whole-cell TREK-2 currents were recorded at +40 mV and perfused with different concentrations of Nb67. Currents were normalised to those recorded prior to the application of nanobody. No activation of TRAAK or TALK-2 K2P channels was observed, but for the related TREK-1 up to 50% inhibition of channel activity was seen at concentrations of Nb67 > 1 μM. (Error bars represent mean ± S.D; $n \geq 3$). **C** Expanded view of the interaction of Nb67 with the two sides of the Cap domain. The interaction with Nb67$_A$ (Left) shows no distortion of the helices, whereas interaction with N67$_B$ on the opposite side results in unwinding of EH1$_B$ (residues E113, E116 and D120 are unique in TREK-2 vs TREK-1). Right: the interaction of N108 with NB67$_B$ differs on the side where EH1$_B$ is unwound. This interaction does not occur with Nb67$_A$.

TRAAK. However, an inhibitory effect was observed on the closely related TREK-1 channel at high concentrations of nanobody >1 μM (Fig. 3B). This opposite effect initially appeared interesting because of the opposite effects of extracellular pH on the activity of TREK-2 and TREK-1, and the role of the extracellular Pore/TM loops in this process[33]. However, Nb-Activator-67 does not appear to contact these loops and instead only interacts with the extracellular face of the Cap domain resulting in an unwinding of the EH1 Cap helix on one side (Fig. 3C). Intriguingly, both M4 helices in this TREK-2 structure are in the Up conformation consistent with a more activated state of the channel (Fig. 3A), although this could also be due to its interactions with the nanobody bound to its intracellular surface. Full occupancy of ion binding sites within the filter of TREK channels is also usually associated with an activated state of the channel[9,34], and in this structure all the sites appear occupied, albeit in an asymmetric fashion due to unwinding of the Cap on one side (Fig. 3A). Although it is tempting to associate the effect of unwinding of EH1 with the activatory effects of Nb-Activator-67, the crystal packing interactions make interpretation of this particular structure challenging. Due to the additional lack of specificity between TREK-1 and TREK-2, the mechanism of activation by this nanobody was not investigated further. However, its robust activatory effect on recombinant homomeric TREK-2 channels, especially at lower concentrations, indicates that this nanobody may remain a useful tool in situations where the identity of its target is already known.

## A selective activatory nanobody

Unlike Nb-Activator-67, the second activatory nanobody (CA10776, hereafter referred to as Nb-Activator-76 or Nb76) exhibited highly selective activation of TREK-2 with an efficacy of 412 ± 14 nM and had no effect on TREK-1 or other K2P channels tested, even when used at high concentration (Fig. 4A & Supplementary Fig. S4C). The structure of this nanobody in complex with TREK-2 reveals extensive contacts between the side of the Cap domain and the CDR loops (Fig. 4B). Interestingly, this involves many residues at the intersubunit interface, several of which are unique in TREK-2 including E113, H135 and Q127. The most significant contact involves W98 on CDR3 of the nanobody which inserts into the intersubunit groove between helices in the Cap domain to form a polar interaction with E128 of TREK-2. Also, R53 on CDR2 helps anchor the nanobody to the Cap helices by H-bonding with Q106 on TREK-2 (Fig. 4C). The importance of these two interactions is supported by the fact mutation of either of these residues on Nb-Activator-76 (W98A or R53A) severely impaired its activatory effect (Fig. 4A), but whether they disrupt binding or nanobody efficacy remains to be determined.

However, the most important functional interactions likely involve contacts with the extracellular P2-M4 loop of TREK-2 that has been implicated in the regulation of filter-gating[33]. In particular, N56 on CDR2 H-bonds with N292 on the P2-M4 loop of TREK-2 (Fig. 4D). The importance of this interaction with the P2-M4 loop is confirmed by mutation of N56 on Nb-Activator-76 which reduces its activatory

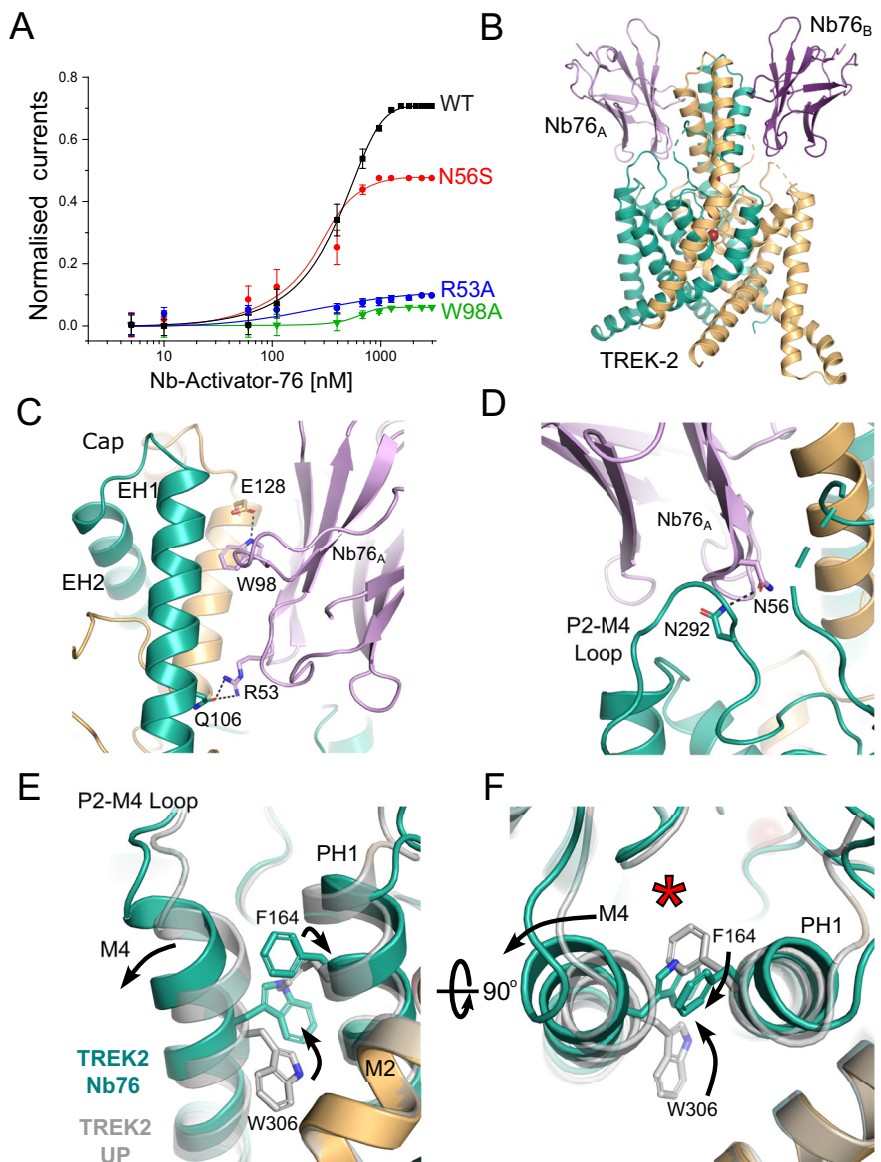

**Fig. 4 | Selective extracellular activation of TREK-2 by Nb-Activator-76. A** Dose-response relationships showing activation of TREK-2 channel activity by Nb-Activator-76 (Nb76). Whole-cell TREK-2 currents were recorded at +40 mV and perfused with different concentrations of Nb76. The effect of mutating different residues in Nb76, which interact with TREK-2, are shown. The curves shown for the R53A and W98A mutant nanobodies were fitted by hand. (Error bars represent mean ± S.D; $n \geq 5$). **B** Crystal structure of TREK-2 in complex with Nb76 bound to both sides of the Cap and the P2-M4 loop. **C** Interactions with the side of the Cap. W98 on CDR3 inserts between Cap helices of different TREK-2 subunits to form an interaction with E128 on EH2 of TREK-2, whilst R53 on Nb76 interacts with Q106 on

EH1 of a different subunit of TREK-2. Mutation of either W98 or R53 on Nb76 reduces its activatory effect. **D** Expanded view of the interaction of N56 on CDR3 with N292 on the P2-M4 loop. Mutation of N56 also reduces activation by Nb76. **E** Alignment with TREK-2 Up-state (4BW5 in Grey) highlights displacement of the top of M4 and the inward rotation of W306. This is accompanied by the outward rotation of F164 on Pore-Helix-1 (PH1). **F** The right-hand panel rotates this view through 90° to better visualise the relative reorientation of W306 and F164 and the outward displacement of M4. This expands the K2P modulator site behind the selectivity filter (red asterisk).

effect, (Fig. 4A). Consistent with an activated state, both M4 helices in this structure are also in the Up conformation (Fig. 4F). The role of ions in gating the filter is unclear because crystals of sufficient quality could only be obtained using a crystallisation solution containing Ba²⁺ which contributes to the crystal contacts. However, this does not change the selectivity filter in comparison to the Up-state conformation (PDB ID 4BW5).

ML335 is a small molecule TREK channel agonist that binds within the 'K2P Modulator Pocket', a cryptic binding site created by the displacement of the top of M4 where its interactions with the P2-M4 loop and Pore-Helix-1 (PH1) are proposed to stabilise TREK-2 in a more conductive conformation[11,17,35]. It is therefore highly significant that the

interaction of Nb-Activator-76 with the P2-M4 loop also induces a displacement in the top of the M4 helix similar to that seen in the ML335-bound structures of TREK-2 (Fig. 4E). Furthermore, it has been reported that the reorientation of a tryptophan residue at the top of M4 (W306 in TREK-2) also correlates with the activated state of TREK channels[8,10,12] and that any hydrophobic moiety occupying this space is capable of stabilising this activated state[10]. In this nanobody bound structure W306 side chains in both helices are in the inward orientation and occupy this space. But unlike other TREK structures where this M4 tryptophan has rotated inwards, here the adjacent residues on PH1 (F164) also flips orientation to interact with the top of M4 to stabilise this displacement (Fig. 4F). This demonstrates a mechanism

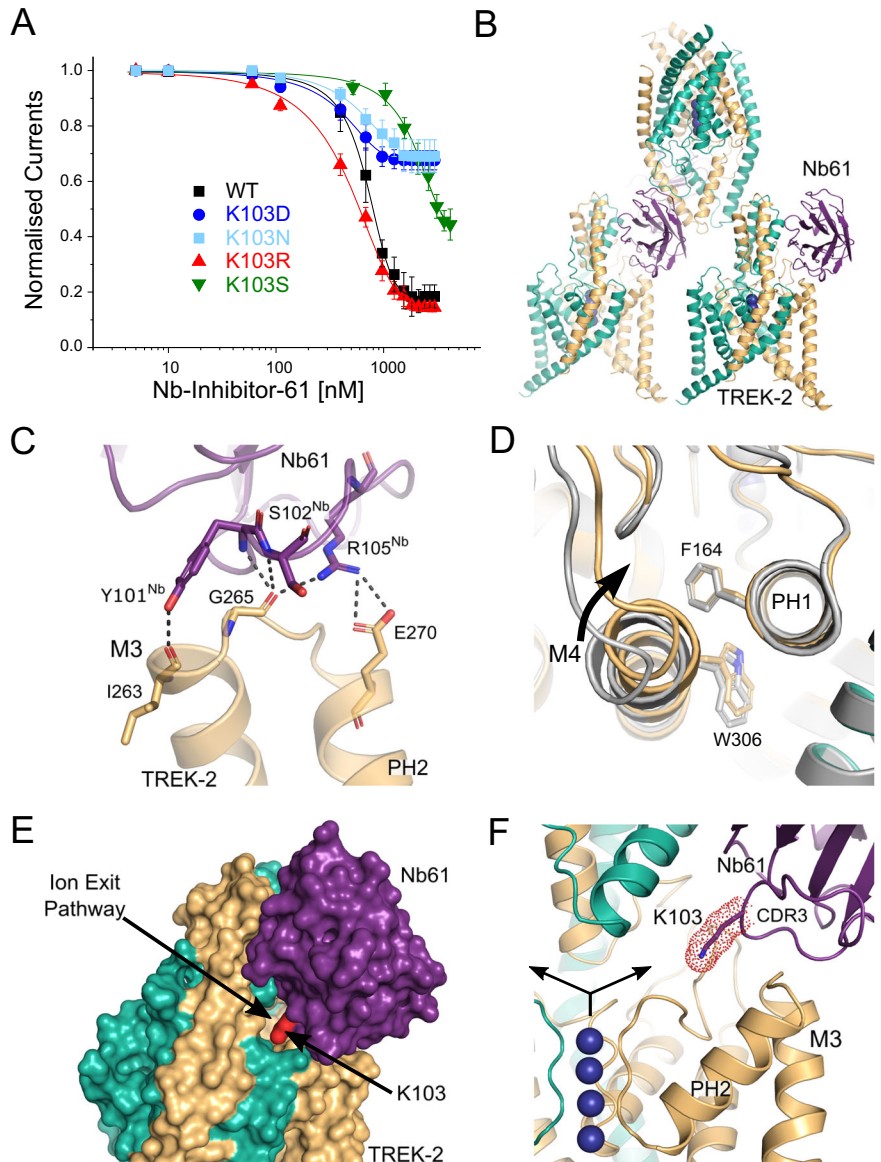

**Fig. 5 | Extracellular inhibition of TREK-2 by Nb-Inhibitor-61. A** Dose-response relationships showing inhibition of TREK-2 channel activity by Nb-Inhibitor-61 (Nb61). Whole-cell TREK-2 currents were recorded at +40 mV and perfused with different concentrations of Nb67. Also shown are the effect of different mutations of K103 in Nb61. (Error bars represent mean ± S.D; $n \geq 5$). **B** The crystal structure unit cell reveals only one Nb61 bound to TREK-2; packing interactions between the Cap domains within the crystal prevent binding to the other side. **C** Expanded view of the major interactions of Nb61 with the top of M3 and the M3-PH2 loop.

**D** Alignment with TREK-2 Up-state (4BW5 in Grey) highlights displacement of M4 and PH2-M4 loop inwards towards the K2P modulator pocket. **E** Surface view showing obstruction of the extracellular ion exit pathway by Nb61 (in purple). K103 within the obstructed pathway is also marked (in red). **F** Expanded view of the location of K103 within the ion exit pathway. K⁺ ions are shown as purple spheres and the arrows indicate the direction of K⁺ flow through the ion exit pathway that is obstructed by K103 on CDR3 of Nb61.

whereby movement of the M4 helix may communicate with PH1 to influence the selectivity filter gate and highlights the importance of this 'K2P modulator pocket' in channel gating[36].

**An inhibitory nanobody with a dual mechanism of action**

The inhibitory nanobody, CA10761, hereafter referred to as Nb-Inhibitor-61 (or Nb61), rapidly inhibited TREK-2 currents ($IC_{50} = 685 \pm 18$ nM; Fig. 5A). This inhibitory effect appears specific as this nanobody has no functional effect on the related TREK-1 or on TASK-3 channels but is active against mouse TREK-2 (Supplementary Fig. S5A). The structure of the nanobody/TREK-2 complex indicates only one nanobody bound per asymmetric unit, but this appears to result from its crystal packing where interactions between the Cap domains prevent binding of a second nanobody to the opposite side (Fig. 5B).

The position of Nb-Inhibitor-61 on the side of the Cap domain allows it to form extensive interactions between the Cap helices and CDR1, as well as with the β5 sheet and β5-β6 loop. In addition, its long CDR3 loop forms multiple interactions with the top of M3 and the M3-PH2 loop (Fig. 5C) as well as additional interactions with the top of M2 and the P2-M4 loop. However, the most striking feature of this long CDR3 loop is that it also inserts a lysine residue (K103) deep within the extracellular ion exit pathway that emerges just below the Cap domain (Fig. 5E, F). Overall, the position of this nanobody obstructs most of this ion exit pathway with the remaining gap being only 1.35 Å wide compared to >4 Å in the *apo* state. Intriguingly, the position of this positively charged lysine within the ion exit pathway is reminiscent of the way some toxins block K⁺ channels via a "cork in a bottle" or "plug" type mechanism[37]. It also shares similarities with the 'finger in the dam'

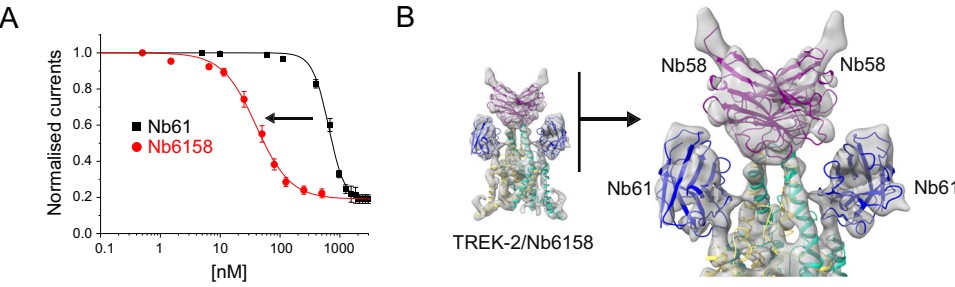

**Fig. 6 | Engineering of a high-affinity biparatropic inhibitory nanobody. A** Dose-response relationships showing inhibition of TREK-2 channel activity by the divalent linked Nb-Inhibitor-6158 (Nb61). Whole-cell TREK-2 currents were recorded at +40 mV and perfused with different concentrations Nb6158. Consistent with its markedly increased affinity, this linked nanobody exhibits a marked increase in its inhibitory efficacy compared to Nb61 alone. (Error bars represent mean ± S.D; $n \geq 5$)

**B** A 7.4 Å resolution CryoEM structure of TREK-2 in complex with Nb6158 shows two copies of Nb61 bound, one to each side of the Cap domain. An expanded view of their relative position is shown on the right. This indicates that, in addition to any allosteric effects, Nb61 is capable of interacting with TREK-2 on both sides of the Cap domain to obstruct $K^+$ permeation from both extracellular ion exit pathways.

mechanism of TREK channel block by Ruthenium Red where interaction of this positively charged inhibitor with the 'keystone inhibitor site' below the Cap domain directly blocks $K^+$ permeation[7,36,38]. Unlike these other toxins, K103 does not insert far enough into the pathway to interact directly with ions exiting the filter, but it clearly has the ability to impede $K^+$ movement through this exit pathway and so we examined its role further.

### A critical role for Lysine 103 in channel inhibition

This lysine appears to make no specific H-bonding or charged interactions within the ion exit pathway and mutation to a serine (K103S) resulted in a marked reduction in its inhibitory effect (Fig. 5A). We also examined other mutations at this position (K103R, K103N and K103D). Consistent with a role for a positively charged residue at this position, the K103R mutant produced similar, if not greater, inhibition whilst neutral or negative substitutions (K103N and K103D) resulted in a markedly reduced inhibitory effect (Fig. 5A). We also examined the effect of this nanobody on TREK-2 single channel activity (Supplementary Fig. S5B). Due to the low intrinsic open probability of TREK-2, only a qualitative comparison was made between the behaviour of multi-channel patches with or without Nb-Inhibitor-61 included in the pipette solution. This revealed a marked suppression of TREK-2 activity via reduction in the durations of bursts of openings and increase in the time spent in the long-closed states. This is consistent with the idea this nanobody acts as a slow blocker resulting in a complete cessation of $K^+$ flux through the channel.

### Possible allosteric effects of Nb-Inhibitor-61

Together, these results with Nb-Inhibitor-61 support the contribution of a direct toxin-like (or 'finger in the dam') block of ion permeation by the nanobody[38]. However, the bifurcation of the ion exit pathway means that such a mechanism would also require nanobody binding to both sides of the Cap domain. There are no obstructions to the binding of a second nanobody on the other side, but due to crystal packing, only one is observed in this structure. It is therefore possible that other, allosteric mechanisms may also contribute to its inhibitory properties. For example, multiple effects are also seen in some toxins where, in addition to direct pore block, interactions with the external pore loops result in allosteric modulation of channel gating[39,40]. For Nb-Inhibitor-61, this may arise through the extensive contacts it makes with the M3-PH2 loop as well as with the P2-M4 loop and M2/M3 helices. Of particular note are the interactions with both I263 and G265 at the top of M3 (Fig. 5C) because E264 has been shown to play an important role in the regulation of the filter gate in TREK channels, thus interaction with the two adjacent residues might easily influence this gating mechanism[11].

Consistent with such an allosteric effect, the M4 helix of the subunit that interacts with this nanobody is in the Down conformation (Fig. 5B). Furthermore, the relative orientation of side chains for both W306 on M4 and F164 on PH1 are reversed compared to their positions the Nb76/TREK-2 structure and its interaction with the P2-M4 loop pulls the top of M4 closer to PH1 compared to the activated state induced by Nb-Activator-76 (Fig. 5D). However, all 4 ion binding sites in the filter appear to be occupied and no obvious structural changes are seen in the filter itself or in the surrounding region. Thus, although additional allosteric effects of this nanobody on the filter gate cannot be excluded, Lysine 103 clearly contributes to channel inhibition by directly blocking the ion exit pathway.

### A biparatopic inhibitory nanobody with improved sensitivity

The two activatory nanobodies identified in this study function with relatively high efficacy ($EC_{50}$-100 and 300 nM) compared to Nb-Inhibitor-61 ($IC_{50}$-650 nM). Multivalency is a well-established method for increasing the affinity of ligands, and linked biparatopic nanobodies which target spatially discrete epitopes have also proven an effective way of increasing their functional efficacy[41–43].

We therefore attempted to increase the sensitivity of Nb-Inhibitor-61 by linking it to the non-functional Cap binder (Nb-Binder-58). Comparison of the TREK-2/Nanobody structures (Fig. 1B) revealed non-overlapping binding sites for these two nanobodies with each nanobody separated ~21 Å between the C-terminus of Nb-Inhibitor-61 and the N-terminus of Nb-Binder-58. The two nanobodies were therefore linked in frame using a $(Gly-Ser)_4$ linker to create a divalent biparatopic nanobody we refer to as Nb-Inhibitor-6158 (or Nb6158).

Analysis of the relative binding affinity of this biparatopic nanobody using bilayer interferometry revealed a marked increase in affinity for TREK-2 of at least 250-fold ($4.1 \pm 0.1$ nM, $n = 4$) compared to monomeric Nb-inhibitor-61 ($1100 \pm 30$ nM, $n = 4$;). Nb6158 also exhibited a 16-fold increase in its inhibitory efficacy from $651 \pm 18$ nM to $40 \pm 3$ nM, $n = 5$ (Fig. 6A). This demonstrates the suitability of linking nanobodies in 'daisy-chain' arrangements to modulate their binding properties.

The structures above show that Nb-Binder-58 binds as a dimer to the Cap domain suggesting that two copies of Nb-Inhibitor-61 in this biparatopic nanobody may now be able to bind more efficiently and directly occlude both ion exit pathways. Given the arrangement of the unit cell in the crystal structure of the TREK-2/Nb61 complex which only shows one Nb61 bound, we attempted to determine its structure in complex with TREK-2 by cryo-electron microscopy (Fig. 6B and Supplementary Fig. S6). At ~7.4 Å resolution, the map was sufficient to reveal two copies of Nb-Inhibitor-6158 bound with one Nb-Inhibitor-61 domain on either side of the Cap domain in an identical position to that

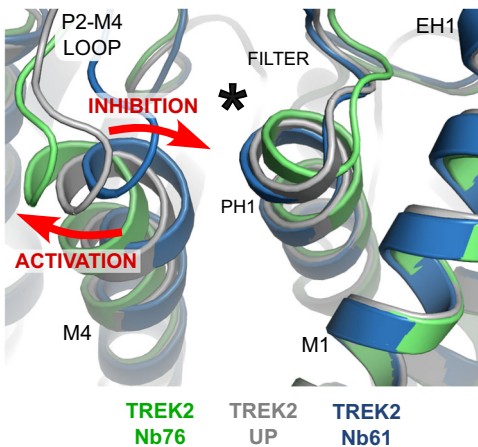

**Fig. 7 | Activation and Inhibition by opposite movement of the top of M4.**
Alignment of the TREK-2/Nb76 and TREK-2/Nb61 complexes with the TREK-2 Up-state show that activation by Nb76 is associated with outward movement of the top of M4 and the P2-M4 loop whilst inhibition by Nb61 involves movement in the opposite direction (red arrows). The location of the K2P modulator site behind the selectivity filter is indicated by the asterisk.

seen in the crystal structure. This confirms that this nanobody can interact with both subunits of TREK-2 and potentially occlude both extracellular ion exit pathways.

In this study, we describe the generation and characterisation of a panel of nanobody binders to the TREK-2 K2P channel resulting in one inhibitor, two activators and other nanobodies suitable for immuno-detection of TREK-2. One activator in particular (Nb-Activator-76) elicits highly selective activation of TREK-2. These nanobodies therefore not only provide useful probes for analysing the structure and function of homomeric TREK-2 channels but also identify a class of highly specific extracellular modulators that can be further investigated for therapeutic potential.

Ever since the first structures of K2P channels were resolved over a decade ago the role of their large extracellular Cap domain has remained unclear[34,44]. It was initially suggested this domain may have evolved to protect the selectivity filter of K2P channels from the small peptide toxins that often target other members of the superfamily of tetrameric cation channels. However, the complex domain swap involving the M1 helix required to assemble this extracellular domain suggests this is an unlikely strategy, especially as examples of toxin resistance in ion channels often only require single mutations in the extracellular toxin binding site[45].

Nevertheless, the Cap domain is a defining feature of K2P channels, and whatever its purpose, it obviously makes the toxin block of the pore much less likely. It is therefore significant that we have identified a nanobody in which the longer CDR3 loop inserts a lysine sidechain directly into the extracellular ion exit pathway in a manner analogous to some K+ channel toxins. In the case of TREK channel inhibition by Ruthenium Red, this small positively charged inhibitor is able to access deeper into the exit pathway to block permeation directly above the filter, whereas it may not be possible for this CDR3 loop to insert deep enough to access the keystone Inhibitor site, even if it were longer. Nevertheless, this inhibitory nanobody is highly effective and its mechanism of action likely involves both allosteric and direct blocking effects on permeation. Also, the structure of this TREK-2/nanobody complex, and those of the others solved in this study, reveal that the Cap domain is highly immunogenic and provides a good scaffold on which nanobodies can anchor themselves to influence TREK channel function. We also demonstrate that the Cap domain allows the design of biparatropic nanobodies with improved affinity and functional efficacy, and the nanomolar efficacy of this

linked Nb-inhibitor-5861 provides a powerful analytical tool for probing TREK-2 function.

In the case of the two activators, the first (Nb-Activator-67) is a highly effective probe for the study of homomeric TREK-2 when the identity of the target is known, but uncertainties about this crystal structure make interpretations of its mechanism of action difficult. However, the second activator (Nb-Activator-76) provides important insights into the mechanism of TREK channel activation and demonstrate how the extracellular P2-M4 loop and rearrangements at the top of the M4 helix play a critical role in the regulation of channel gating. Overall, these rearrangements expand current models for TREK channel gating via the K2P modulator site to suggest that outward displacement of the top of M4 changes the contacts between M4, the P2-M4 loop and Pore-Helix-1 to activate the selectivity filter gate, whereas displacement in the other direction results in the opposite effect and inhibition. Similar to that proposed for other TREK channel modulators[8,10,17], this may also involve reorientation of the W306 and F164 side chains, or their equivalents (Fig. 7). This model is also consistent with recent observations that covalent attachment of ML335 derivatives within the K2P modulator site via a conserved serine on PH1 can irreversibly activate TREK channels[35].

However, the mechanism by which nanobody binding is then coupled to changes in ion occupancy within the filter and the dynamics of K+ permeation remain unclear. A limitation of this study is that the relative dynamics of these specific regions are probably critical for the regulation of gating, but most structures, including those in this study, are determined in high [K+] (>100 mM). This often results in high ion occupancy within the filter in such crystal structures, whereas the physiological situation is obviously very different and difficult to compare. It also remains to be determined how these modulatory nanobodies interact with other regulatory mechanisms such as temperature and stretch. However, the cooperative and polymodal nature TREK channel gating means any such interactions are likely to be complex and potentially difficult to dissect.

The nanobody 'binders' that have no functional effects are also likely to be useful tools for the immunodetection of TREK-2, although future studies will be required to demonstrate the sensitivity required for detection in vivo. It is also known that TREK-2 can form heteromeric channels with either TREK-1 or TRAAK[46,47], and nanobodies capable of detecting such combinations would be extremely powerful for dissecting the role of such heteromeric channels in vivo. It is therefore significant that the domain swap involved in the assembly of the Cap domain generates an intersubunit interface that forms extensive contacts with nanobodies in these structures and which would be unique in any heteromeric channel combinations; thus, if binders can differentiate between these interfaces in homomeric TREK channels, then it may be possible to generate similar nanobodies that are uniquely selective for heteromeric K2P channels.

Thus, although ion channels remain underexploited as antibody drug targets, this study demonstrates that the unique nature of the Cap domain combined with the role of the extracellular loops in K2P channel gating allows nanobodies to be considered as an exciting alternative to small molecules as highly selective probes to K2P channel structure-function studies, and of TREK channels in particular. This therefore expands the potential of nanobodies as effective therapeutics and their role in future drug development.

## Methods

### Nanobody generation, expression and purification
Nanobodies were generated as part of the Nanobodies4Instruct program using the method described by Pardon et al. (28). Briefly, detergent-solubilised purified human TREK-2 protein (Gly67 to Glu340), identical to that used in previous structural studies (9) was used to elicit an immune response in llamas. The resulting immune repertoire was isolated and cloned into the pMESy4 phage display

vector. Positive clones were selected by phage panning and screening against the same TREK-2 protein. A total of 29 unique nanobody clones were identified. The sequence alignment shown in Supplementary Fig. S1 are highlighted using TEXshade. To create the biparatropic Nb6158, extension overlap PCR was used to link the C terminus of Nb61 to the N-terminus of Nb58 with a $(GGGGS)_3$ linker. Periplasmic expression and purification were modified from Pardon et al. [31]. Briefly, relevant nanobodies in the pMESy4 vector which has a N terminal PelB signal sequence and a C terminal 6x-His tag. These were transformed into the WK6 strain of *E. coli*. 10 ml overnight cultures were then used to inoculate master cultures (1 L Terrific Broth, 4 ml Glycerol, 0.1% w:v glucose). These were then grown at 37 °C shaking at 180 rpm with expression induced by 0.5 mM IPTG when $OD_{600}$ reached 0.8. Growth was then continued overnight at 28 °C. Cells were harvested by spinning at 5000 g for 20 min at 4 °C and the pellets resuspended in ice-cold resuspension buffer (in mM: 200 Tris, 0.5 EDTA, 500 Sucrose). To disrupt the periplasm, cells were kept on ice and shaken for 1 h in resuspension buffer, followed by a rapid 3-fold dilution with a 5 mM $MgCl_2$ solution and then shaken for a further hour. The periplasm was then recovered by spinning at $15000 \times g$, for 15 min at 4 °C. His-tagged nanobodies were then purified by the addition of Ni-NTA resin to the supernatant and incubating at 4 °C for 2.5 h. Following washing with Wash Buffer 1 (in mM: 50 Sodium Phosphate, 1000 NaCl, pH7.0) and Wash Buffer 2 (in mM: 50 Sodium Phosphate, 1000 NaCl, pH6.0), nanobodies were eluted in Elution buffer (in mM: 50 Sodium Phosphate, 150 NaCl, 300 Imidazole, pH 7.0), and the imidazole rich solution exchanged into either Gel Filtration Buffer (200 mM KCl 20 mM HEPES pH7.5, for screening experiments), ND96 (in mM: 96 NaCl, 2 KCl, 1.8 $CaCl_2$, 1 $MgCl_2$, 20 HEPES, 2.5 Na Pyruvate, for Two-Electrode Voltage Clamp studies), or extracellular buffer (in mM: 140 NaCl, 10 TEA-Cl, 5 KCl, 3 $MgCl_2$, 1 $CaCl_2$, 10 HEPES, pH 7.4, for patch clamp electrophysiology) using PD10 columns. Purity was assessed by SDS-PAGE. An initial assessment of nanobody binding to TREK-2 was done with size exclusion chromatography. Each nanobody was mixed with purified TREK-2 in a 2:1 molar ratio and applied to a Sepax SRT SEC-300 column, pre-equilibrated in Gel Filtration buffer. Out of the 29 initial nanobodies, 16 co-eluted with TREK-2 and produced a visible shift of the 280 nm TREK-2 peak, compared to TREK-2 injected without nanobody.

## Crystallography and data collection
Human TREK-2 protein (residues Gly67 to Glu340), identical to that used in our recent structural studies was expressed and purified as previously described[4,9] and relevant nanobodies were prepared as described above. For Nb67 and Nb61 a TREK-2 glycosylation mutant (TREK-2[glyco]) was used; this was purified in an identical manner and contained the following substitutions (N149Q, N152Q and N153Q; none of these sites form contacts with the nanobody). TREK-2 protein was mixed with the nanobody in a 1:2 TREK-2:nanobody molar ratio prior to crystallisation. Crystals formed in vapour diffusion drops (TREK-2-Nb58: 0.1 M glycine pH 10.0, 0.2 M NaCl, 31% v/v polyethylene glycol (PEG) 1000; TREK-2[glyco]-Nb67: 0.1 M Tris pH 8.0, 65% v/v 2-methyl-2,4-pentanediol; TREK-2-Nb76: 0.1 M HEPES pH 7.0, 0.3 M $BaCl_2$, 39% PEG 400), apart from the TREK-2[glyco]-Nb61 complex crystals, which grew in Lipidic Cubic Phase (MAG7.9 with 3:2 lipid:protein ratio, 0.1 M Na acetate pH 4.5, 0.3 M KCl, 23% v/v PEG 400). Vapour diffusion crystals were grown at 4 °C and LCP crystals at 20 °C for 1–12 weeks. Crystals were screened, and X-ray diffraction data collected on beamline I24 (Diamond Light Source) at 100 K. Data were recorded at a wavelength of 0.9686 Å on a Pilatus3 6 M detector from single crystals using 0.1 or 0.2° oscillation and a beam-size of 20 × 20 μm (30 × 30 μm for TREK-2-Nb76). Data were processed, reduced and scaled using XDS[48] and AIMLESS[49]. All datasets exhibited moderate to strong diffraction anisotropy and were further processed using STARANISO to apply an anisotropic diffraction cut-off to the merged data as well as structure

amplitude estimation and anisotropic correction (see Supplementary Table S1 for resolution limits of data and completeness). Structures were phased by molecular replacement using the 4BW5 (TREK-2 [glyco]-Nb67 and TREK-2-Nb76), 4XDJ (TREK-2 [glyco]-Nb61) or 4XDK (TREK-2-Nb58) structures as starting models for TREK-2[9] and 5HVG (Nb58 and Nb76), 5UKB (Nb67) or 5IMK (Nb61) for the nanobodies. Structures were manually built with COOT and refined with BUSTER (v2.10.4). Models were refined with LSSR restraints[50] and a single translation/libration/screw model group was refined for each protein chain.

## Cryo-electron microscopy and data collection
Purified TREK-2 was complexed with purified linked Nb6158 at a 1:1.2 molar ratio and incubated for 2 h on ice. TREK-2-Nb complex was separated from free Nb by gel filtration on a Superose 6 Increase column in 200 mM KCl, 20 mM HEPES pH 7.5 and 0.12% OGNG with 0.012% CHS. The complex was concentrated to 5.1 mg/ml and a 3 μl aliquot applied to a glow-discharged Quantifoil holey carbon grid (1.2/1.3 300 Cu mesh) and blotted for 3 s (100% humidity, 4 °C) using a VEI Vitrobot IV. Data was collected at the Central Oxford Structural Molecular Imaging Centre (COSMIC) on a FEI Arctica 200 kV cryo-TEM with FEI Falcon 4 direct electron detector. Movies were recorded at 150000× magnification with a pixel size of 0.94 Å per pixel and a total dose of 30 e-/Å². The defocus values ranged from −3.0 to −1.5 μm. Data was analysed using SIMPLE and CryoSPARC[51,52]. In SIMPLE, manual picking was used to create 2D cluster templates to pick 301,660 particles from 2651 micrographs. The particles were exported to CryoSPARC and following two rounds of 2D classification and selection, 17,788 particles were selected for ab-initio reconstruction followed by homogeneous refinement to give a density map of 7.37 Å. A model was created by overlaying the crystal structures of TREK-2 with Nb61 and Nb58. These were fitted in to the map in COOT[53] and further refined with PHENIX[54].

## Biolayer interferometry
This was performed using an Octet Red385 (Sartorius). Streptavidin-coated biosensors were loaded with biotinylated TREK-2 at 50 nM in 200 mM KCl, 20 mM HEPES pH 7.5 and 0.12% OGNG with 0.012% CHS. To calculate $K_d$, serial dilutions of 2 μM–7.8 nM for Nb 61 and 1000 nM–3.9 nM for Nb6158 were prepared. After a 300 s baseline step, the nanobodies were allowed to associated for 600 s and then dissociate for 600 s. Data was analysed in the Octet v9.0 software package and all raw data were filtered with a Savitzky-Golay filter, baseline and reference subtracted, in-step corrected and y-axis aligned.

## Immunodetection of TREK-2
TREK-2 and other human K2P channels were tagged with mCherry by removal of the stop codon and in-frame fusion of mCherry to the C-terminus using a $(GS)_2$ linker and then cloned into the pFAW vector suitable for mammalian expression. For those K2Ps with a predominantly intracellular distribution trafficking mutations were used to promote surface membrane expression i.e. TWIK1 I293A/I294A[55] and THIK2 ΔN6-23[56]. Nanobodies were tagged by in-frame fusion of mVenus at the C-terminus using a $(GS)_2$ linker, and also then cloned into the pFAW vector. HEK293T cells were grown in 24 wells plates treated with poly-L-lysine and transfected with 250 ng of K2P-mCherry-pFAW as described in the TREK-2-mCherry expression trial. After 30 h, the media aspirated off and the cells fixed using 4% formaldehyde for 10 min and then washed 3× with PBS. The cells were incubated with 150 μl nanobody for 2 h at 37 °C, then washed 4× with PBS before imaging on a Nikon eclipse TE2000U inverted microscope with a GFP and mCherry filter cube. Images were captured on a ZWO ASI 120MC-S camera.

## Electrophysiology
For the microinjection of oocytes, mRNA was in vitro transcribed from the wild-type human TREK-2 gene (*KCNK10*) in the pFAW oocyte

expression vector. Each oocyte was injected with -0.2 ng of RNA and Two-Electrode Voltage Clamp recordings then performed as previously described[13,14]. Briefly, after injection of mRNA, oocytes were incubated for 20–24 h at 17.5 °C and measured in ND96 buffer at pH 7.4 (96 mM NaCl, 2 mM KCl, 2 mM MgCl2, 1.8 mM CaCl$_2$, 5 mM HEPES). Unless otherwise stated, for the dose-response measurements, oocytes were perfused with increasing concentrations of purified nanobody in ND96 solution and currents recorded using a 400 ms voltage step protocol from a holding potential of −80 mV delivered in 10 mV increments between −120 mV and +50 mV and 800 ms ramp protocols from −120 to +50 mV. Unless otherwise described, all results shown are reported as mean ± standard deviation and obtained with oocytes from at least 3 independent replicates. Where individual data points are not shown then $n \geq 5$. For the initial screening experiments nanobody was pipetted directly into the bath chamber in gel filtration buffer and current recorded using a 800 ms ramp from −50 to +80 mV. For the measurement of intracellular effects, nanobodies were applied to the inside of giant excised inside-out membrane patches. The intracellular solution had the following composition (in mM): 120 KCl, 10 HEPES, 2 EGTA and 1 Pyrophosphate (pH adjusted with KOH/HCl) with a pipette solution (in mM): 120 KCl, 10 HEPES and 3.6 CaCl$_2$ (pH 7.4 adjusted with KOH/HCl). The recording of TREK-2 channel activity was then performed as previously described[4]. For single channel recordings, the pipette solution contained (in mM) 116 NMDG, 4 KCl, 1 MgCl$_2$, 3.6 CaCl$_2$, 10 HEPES (pH 7.4); whilst bath solution contained (in mM) 120 KCl, 1 NaCl, 2 EGTA, 10 HEPES (pH 7.3). The currents were filtered at 1 kHz and sampled at 10 kHz.

### Reporting summary
Further information on research design is available in the Nature Portfolio Reporting Summary linked to this article.

## Data availability
All data within this study are included in the article and/or the Supplementary Information, and materials and underlying data are available upon request. The atomic coordinates and structure factors for the nanobody bound complexes of TREK-2 have been deposited in the Protein Data Bank under accession codes: 8QZ1 [(Nb58), 8QZ2 (Nb61), 8QZ3 (Nb67), 8QZ4 (Nb76)]. Electron density for the biparatropic Nb6158 is deposited at the Electron Microscopy Data Bank with the entry ID code, EMD-19066 (Nb6158). Coordinates for the other TREK-2 Up-states structure referenced in the study can also be accessed via the Protein Data Bank: 4BW5. A Source Data File accompanies this manuscript. Source data are provided with this paper.

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

## Acknowledgements

We acknowledge the support and use of resources of Instruct-ERIC (PID1212 VID1054), part of the European Strategy Forum on Research Infrastructures (ESFRI) and the Research Foundation - Flanders (FWO) for support with nanobody discovery and thank Nele Buys for the technical assistance during Nanobody discovery. We also thank Diamond Light Source for beam time (BAG proposal mx15433/mx19301) and the I24 beamline staff for assistance with crystal screening and data collection, as well as the staff of the Central Oxford Structural Molecular Imaging Centre (COSMIC) for assistance with CryoEM. This work was supported by grants from the Biotechnology and Biological Sciences Research Council and Medical Research Council to S.J.T. (BB/T002018/1, BB/S008608/1 and MR/W017741/1) and from the Deutsche Forschungsgemeinschaft to M.S. (SCHE 2112/1-2) and T.B (BA 1793/6-2) as part of the Research Unit FOR2518, DynIon. E.P.C. was supported by the Structural Genomics Consortium, a registered charity (number 1097737) that received funds from AbbVie, Bayer, Boehringer Ingelheim, Genome Canada, Janssen, Merck, Novartis, the Ontario Ministry of Economic Development, Pfizer, and Takeda, as well as the Wellcome Trust (106169/Z/14/Z). S.J.T. and E.P.C. also received support from the Wellcome Trust as part of the OXION Initiatives in Ion Channels and Membrane Transport in Health and Disease (WT084655MA and 102161/B/13/Z).

## Author contributions

S.J.T., E.P.C. and K.E.J.R. conceived/designed the principal elements of the study. All authors (K.E.J.R., A.C., J.St., A.B., K.H.M.S., A.C.W.P., J.A., P.P., M.S., I.H.-K., S.R.B., J.E., E.P., T.B., R.J.O., S.N., J.Sö., E.P.C., and S.J.T.) generated, analyzed or interpreted data or generated materials. E.P. and J.St. (VIB-VUB, Brussels, Belgium) generated nanobody reagents. S.J.T., A.C., J.Sö. and K.E.J.R. (University of Oxford) drafted the manuscript and all authors contributed to the final version.

## Competing interests

The authors declare no competing interests.
