## [Peer Review File · Nature Communications]

Extracellular modulation of TREK-2 activity with nanobodies provides insight into the mechanisms of K2P channel regulationReviewer #1 (Remarks to the Author):

In this manuscript, Rödström et al. provide an elegant answer to the need to develop drugs specific for two-pore domain potassium channels (K2P), a family of channels controlling cellular excitability. They isolated nanobodies generated in llamas after injection of purified TREK2 channel under the same conditions used for its crystallization. They show that more than half of the nanobodies generated can form a complex with the channel, and that 3 of them modify the channel's activity. While two nanobodies activate TREK2, the third appears to be inhibitory. By combining different techniques in which they are recognized specialists - nanobody production, crystallization, electrophysiology and cell biology - the authors demonstrate the effectiveness of combining these skills to study K2P channels and propose nanobodies as a potential new therapeutic tool in disorders linked to K2P channel dysfunction. The inhibitor's identified binding site highlights the importance of a region close to the selectivity filter.

However, while this approach appears very promising and deserves to be published, there are some points in the manuscript that deserve improvement.

1- To demonstrate the effectiveness of nanobodies as a tool for immunodetection of the target channel on the cell surface, the authors use the argument that, in the various resolved structures, the nanobodies bind well to the extracellular CAP domain of TREK2, and that cell-surface detection is therefore possible with these candidates. This argument alone is not sufficient, and systematic immunoassay labellings should be performed under native conditions to be able to affirm that it applies to all the candidates studied here. In addition, the only immunoassay presented, involving the use of two nanobodies that do not change channel activity, lacks methodological clarity. This assay is presented as proof of the specificity of these nanobodies on TREK2 (Supplementary FigureS3). Two criticisms can be made of this assertion and of the presentation of these nanobodies as immunodetection tools: (a) Among the other K2P channels chosen to demonstrate nanobody specificity, some such as TWIK1, TWIK2 or THIK2, display a predominantly intracellular distribution. Nowhere in the manuscript there is any mention of whether or not permeabilization conditions are used. This point about the physical accessibility of nanobodies to their targets needs to be clarified before specificity can be asserted. (b) The authors do not discuss the impact of the fusion of mVenus protein (230 aa) could have on a nanobody (150 aa), even if they are separated by a linker. The statement that mVenus fusion does not change the structure of the nanobody-channel complex should be demonstrated.

The authors must also ensure that the quality of the immunoassay figures presented in the revised version is of optimum quality to allow objective assessment of their conclusions, which is not the case on the pdf provided.

2- The authors show clearly that the activating nanobodies Nb67 and Nb76 display partial and total selectivity for TREK2, respectively, compared with TREK1 which shares good sequence conservation with TREK2. This specificity was not discussed further, and this point was not even investigated for the inhibitory nanobody Nb61. For example, in the case of Nb76, they report that its activating effect on TREK2 is dependent on the interaction of the nanobody's CDR3 and CDR2 with TREK2 residues E128 and Q106 respectively. These two residues are highly conserved in TREK1, whereas Nb76 nanobody has no effect on this channel. The authors did not discuss this point. Similarly, in the case of Nb61, they describe an inhibition of TREK2 linked to the binding of CDR3 residue K103 in a region of the channel for the ion exit pathway, and to the importance of the positive charge of this residue in this region. The authors didn't mention the fact that in this region of the channel is also located residue E264 of M3 which has been shown to be involved in TREK1 C-type gating (E234) in a mechanism conserved in other K2P channels (ref Lolicato 2020). An assessment of the selectivity of Nb61 on TREK2 might have revealed a common mechanism here too. This lack of systematic comparison between the capacity of the nanobodies studied here and other K2P channels becomes frustrating when reading the discussion, which mentions several more or less selective TREK channel modulators such as Ruthenium Red, inactive on TREK1, or M335, inactive on TRAAK.

Nevertheless, the discussion is very clear and objective on the weaknesses of the method, notably with regard to potassium occupancy of binding sites in the selectivity filter, and the fact that electrophysiology is carried out under conditions of potassium ion concentration very different from

those of crystallization. The mention of the interest of this approach for the study of K2P heteromers of the TREK subfamily lacks, however, important bibliographical references on this heteromerization (line 371). Finally, it would have been interesting to gather experimental data on the impact of these nanobodies on other regulations of this TREK2 channel (pH, stretch, T°, ...), or at least to discuss this point.

Minor points :

- 1- A bibliographic reference on the movement of the M4 domain in TREK gating is missing (line 46).
- 2- Line 51, reference 13 concerns an article deposited on BioRxiv in 2021. Unless we are mistaken, this work has since been revised and published under a different title and reference in the same year. Authors should use the latter reference, unless the subject of the quotation does not appear in the published article, in which case they should qualify their statement. The same applies to reference 14 (line 53). This publication has still not been peer-reviewed and therefore cannot be used as a source of evidence.
- 3- Line 161, the concentration noted (10 µm) is different from that used in supplementary Figure S4B (1 µM).
- 4- Line 244: please remove the article "the".
- 5- Figure 2A: Panel A is supposed to show Nb58 in complex with TREK2, but in the right panel the nanobody is labelled N57Nb.3
- 6- Figure 4- Figure 5 legend: line 604, change Nb67 by Nb61.5
- 7- Figure 5C: neither P2-M4 nor M2, both mentioned in the legend (line 607-608), appear in the proposed figure.
- 8- Figure 5D: this panel is supposed to show a view of interactions between N61 and the CAP side, but here the nanobody is labelled N57Nb. Figure 5D: this panel is supposed to represent a view of the interactions between N61 and the CAP side, but it shows a conformational change of M4 with and without the nanobody, which is not shown in this view. To be modified.
- 9- Supplementary Figure S1: The figure is in negativ... (probably an editing problem) and therefore impossible to revise.
- 10- Supplementary Figure S2: The choice of colors to represent the different structures and electron densities is not very judicious and does not make it easy to distinguish nanobody and channel. The legend should explain the significance of the indices A, B, C and D assigned to the various residues shown.
- 11- Supplementary Figure s3 : Image quality is very poor. It is essential to improve the quality so that the reader can properly appreciate the relevance of the conclusion.

Reviewer #2 (Remarks to the Author):

K channels are important drug targets for many diseases but obtaining high selectivity with small molecules can be challenging. This study from Rodstrom et al instead studies nanobodies as alternatives to small molecules, specifically against TREK-2 channels, to create advanced pharmacological tools that could be used both for basic discovery research and explored for therapeutic potential.

Rodstrom et al used purified TREK-2 protein injected into llamas to raise a panel of nanobody binders. From the panel four nanobodies of interest were explored in more detail, one neutral, one inhibitor and two activators. The study uses X-ray crystallography and cryo-EM to reveal the binding modes of each nanobody and provide insights into molecular mechanisms of action. This is supported by a thorough electrophysiological characterisation of function in combination with mutagenesis studies to confirm the roles of residues.

Overall, this study combines a variety of techniques to identify important new pharmacological tools for potassium channel modulation whilst explaining molecular mechanisms of action. It has broad conceptual relevance to the ion channels field for understanding antibody modulation and informing on the advancement of antibody modulators, making it suitable for publication in Nature Comms.

Minor comments are:

- Line 196: What kind of interaction? Polar? Between which parts? Side chains? Backbones?
- Line 201: What kind of interaction? Polar? Between which parts? Side chains? Backbones?
- The "Selective activatory nanobody" section. It would be nice to provide an additional figure pinpointing any residues involved in interactions that differ between TREK-1 and 2, to explain the molecular basis of selectivity. I do not suggest any need for mutagenesis studies to verify this, but inclusion of a figure would be insightful. If there is no clear molecular explanation then it would still be useful and interesting to discuss this.
- The "inhibitory nanobody with dual effects" section. I did not understand why the word "dual" was present. So far as I can tell the nanobody is an inhibitor and does not have dual effects.
- The "A biparatropic inhibitory nanobody with improved efficacy" section. As I understand efficacy it refers to strength of maximal effect, e.g. does an inhibitory nanobody at saturating concentration only inhibit channels by 20% or 80 %, etc? Whereas what you are looking at is sensitivity of effect – concentration required to achieve 50 % inhibition. To avoid confusion in the field replace efficacy with sensitivity.

Dr Paul Steven Miller

NCOMMS-23-60305 RESPONSE TO REVIEWER COMMENTS

We thank the reviewers for their insightful and helpful comments in evaluating this manuscript. We believe we can address all these comments as indicated below and hope that the improved manuscript is now suitable for publication.

Reviewer #1 (Remarks to the Author):

In this manuscript, Rödström et al. provide an elegant answer to the need to develop drugs specific for two-pore domain potassium channels (K2P), a family of channels controlling cellular excitability. They isolated nanobodies generated in llamas after injection of purified TREK2 channel under the same conditions used for its crystallization. They show that more than half of the nanobodies generated can form a complex with the channel, and that 3 of them modify the channel's activity. While two nanobodies activate TREK2, the third appears to be inhibitory. By combining different techniques in which they are recognized specialists - nanobody production, crystallization, electrophysiology and cell biology - the authors demonstrate the effectiveness of combining these skills to study K2P channels and propose nanobodies as a potential new therapeutic tool in disorders linked to K2P channel dysfunction. The inhibitor's identified binding site highlights the importance of a region close to the selectivity filter.

However, while this approach appears very promising and deserves to be published, there are some points in the manuscript that deserve improvement.

1- To demonstrate the effectiveness of nanobodies as a tool for immunodetection of the target channel on the cell surface, the authors use the argument that, in the various resolved structures, the nanobodies bind well to the extracellular CAP domain of TREK2, and that cell-surface detection is therefore possible with these candidates. This argument alone is not sufficient, and systematic immunoassay labelling should be performed under native conditions to be able to affirm that it applies to all the candidates studied here. In addition, the only immunoassay presented, involving the use of two nanobodies that do not change channel activity, lacks methodological clarity. This assay is presented as proof of the specificity of these nanobodies on TREK2 (Supplementary FigureS3). Two criticisms can be made of this assertion and of the presentation of these nanobodies as immunodetection tools: (a) Among the other K2P channels chosen to demonstrate nanobody specificity, some such as TWIK1, TWIK2 or THIK2, display a predominantly intracellular distribution. Nowhere in the manuscript there is any mention of whether or not permeabilization conditions are used. This point about the physical accessibility of nanobodies to their targets needs to be clarified before specificity can be asserted. (b) The authors do not discuss the impact of the fusion of mVenus protein (230 aa) could have on a nanobody (150 aa), even if they are separated by a linker. The statement that mVenus fusion does not change the structure of the nanobody-channel complex should be demonstrated.

a) The potential of these Nbs for immunodetection only forms a relatively small part of this larger study. However, we agree that the methodological description was not clear enough and have now included more details (the cells were fixed, but not permeabilised). Also, in the case of the channels quoted which display mostly intracellular distribution (TWIK/THIK), we used mutations shown to traffic these channels to the plasma membrane (i.e. I263A/I264A in TWIK 1 and Δ N6-23 in THIK2) - this is also now indicated in the methods. Combined with the fact there is no signal with Nbs57&58 from the highly related TREK1 (Fig 2B) we believe it is fair to conclude that these Nbs exhibit an extremely high degree of specificity for TREK2 under the conditions used here, and may therefore be considered useful tools for others to examine under native conditions – we consider such expts in native tissues beyond the remit of this study and only present immunoassay data for the non-functional nanobodies (Nb57 and Nb58) as we believe these are likely to be the most useful Nbs for future exploitation in any such immunodetection experiments.

b) With regards the second query about whether the mVenus fusion alters the nanobody/channel complex - the fusion was made at the C-terminus of Nb58 and we have now indicated the position of the C-termini in Fig2A. This shows that this region is located on the opposite side to any of the Nb/channel interfaces and so is extremely unlikely to influence the Nb/channel complex. The fact this

Nb fusion also still binds to its intended target with great specificity in our immunoassay strongly supports this conclusion. Furthermore, as shown in the CryoEM structure in Fig. 6, fusion of Nb61 to the N-terminus has no effect on binding of Nb58. Also, the C-terminus is even further away from the Nb/Channel interface, but the only way to determine this definitively is to resolve another CryoEM structure which is not an efficient use of limited time/resources given the results of the immunoassay.

The authors must also ensure that the quality of the Immunoassay figures presented In the revised version is of optimum quality to allow objective assessment of their conclusions, which is not the case on the pdf provided.

We apologise for the quality – initial submission was via pdf which is then reconverted resulting in a loss of quality – especially with the supplementary Fig S3. Higher quality versions of Fig2 are now uploaded with the resubmission, but the majority of the panels in Fig S3 show negative data i.e., empty black boxes for the other K2Ps shown – the positive results are shown in the main Fig 2B and are clearer to see. We have increased the resolution of the supplementary Figure so that it can be magnified for viewing, but the pdf is now quite large and we cannot increase it further – we would rather not try to increase the brightness artificially. Our conclusions about the suitability of these Nbs for immunodetection only form a relatively small part of this study and we believe the data we present is commensurate with the conclusions we make and will allow others to follow up with more detailed studies if they wish – Gold standard validation ultimately requires immunostaining of WT vs knockout mouse tissues which are beyond the remit of this initial study. Upon publication these Nb clones will be made freely available for such further studies and future engineering for conjugation to brighter dyes or secondary antibody detection etc.

2- The authors show clearly that the activating nanobodies Nb67 and Nb76 display partial and total selectivity for TREK2, respectively, compared with TREK1 which shares good sequence conservation with TREK2. This specificity was not discussed further, and this point was not even investigated for the inhibitory nanobody Nb61.

We thank the reviewer for spotting that we omitted to show specificity data for Nb61. This is now included as Fig S5A - it has no effect on TREK1, or TASK3 channel activity – the figure also shows that it activates mouse TREK2 which is very highly conserved with human TREK2.

For example, in the case of Nb76, they report that its activating effect on TREK2 is dependent on the interaction of the nanobody's CDR3 and CDR2 with TREK2 residues E128 and Q106 respectively. These two residues are highly conserved in TREK1, whereas Nb76 nanobody has no effect on this channel. The authors did not discuss this point.

We apologise for the lack of clarity here. These are indeed two residues which form contacts at the side of the Cap domain and were amongst the most obvious to represent examine, but there are other TREK2/Nb interactions which are unique and which were not discussed (Q113/R100, H135/T33 & Q127/L47) – these are now mentioned in the text. However, the complexity of these multiple interactions at this interface make visualization in a figure challenging. We also qualify our results by mentioning that we do not know whether disruption of the two interactions mutated abolishes Nb binding. We propose that the functionally relevant interaction is with the P2-M4 loop as shown in Fig.4D and we now clarify this within the text. Please note we specifically avoided mutating residues within TREK2 itself to examine the functional relevance of these interactions because any residues that affect the assembly/trafficking/gating of TREK2 itself would likely complicate meaningful interpretation of the results.

Similarly, in the case of Nd61, they describe an inhibition of TREK2 linked to the binding of CDR3 residue K103 in a region of the channel for the ion exit pathway, and to the importance of the positive charge of this residue in this region. The authors didn't mention the fact that in this region of the channel is also located residue E264 of M3 which has been shown to be involved in TREK1 C-type gating (E234) in a mechanism conserved in other K2P channels (ref Lolicato 2020). An assessment of the selectivity of Nb61 on TREK2 might have revealed a common mechanism here too. This lack of systematic comparison between the capacity of the nanobodies studied here and other K2P channels

becomes frustrating when reading the discussion, which mentions several more or less selective TREK channel modulators such as Ruthenium Red, inactive on TREK1, or M335, inactive on TRAAK.

We agree with this comment – even though we discuss at length that this Nb probably has additional allosteric effects, we only briefly mentioned the interactions with the top of M3, and agree we missed an opportunity to cite the work of Lolicato *et al* and the role of E264 in channel gating. This is now discussed in more detail in this section and referenced appropriately.

Nevertheless, the discussion is very clear and objective on the weaknesses of the method, notably with regard to potassium occupancy of binding sites in the selectivity filter, and the fact that electrophysiology is carried out under conditions of potassium ion concentration very different from those of crystallization. The mention of the interest of this approach for the study of K2P heteromers of the TREK subfamily lacks, however, important bibliographical references on this heteromerization (line 371). Finally, it would have been interesting to gather experimental data on the impact of these nanobodies on other regulations of this TREK2 channel (pH, stretch, T°, ...), or at least to discuss this point.

We have now included appropriate bibliographic references to the important role of heteromeric TREK1/TREK2 channels. We have now added mention of other activatory modalities in the discussion, but have not examined the effect of these Nbs in combination with activatory mechanisms such as stretch and temperature as any such results would likely be difficult to interpret due to the highly cooperative and polymodal mechanisms of gating in TREK channels.

Minor points :

1- A bibliographic reference on the movement of the M4 domain in TREK gating is missing (line 46).

Corrected

2- Line 51, reference 13 concerns an article deposited on BioRxiv in 2021. Unless we are mistaken, this work has since been revised and published under a different title and reference in the same year. Authors should use the latter reference, unless the subject of the quotation does not appear in the published article, in which case they should qualify their statement. The same applies to reference 14 (line 53). This publication has still not been peer-reviewed and therefore cannot be used as a source of evidence.

This first reference is now updated to the published paper. It is now accepted practice in all major journals to reference *bioRxiv* papers as sources of evidence where appropriate. The paper in question is under revision with a major journal and the relevant *bioRxiv* link will be updated in time with a live link to the final article.

3- Line 161, the concentration noted (10 μ m) is different from that used in supplementary Figure S4B (1 μ M).

Now corrected.

4- Line 244: please remove the article "the".

Done

5- Figure 2A: Panel A is supposed to show Nb58 in complex with TREK2, but in the right panel the nanobody is labelled N57Nb.3

Thanks for spotting this - mistake corrected

6- Figure 4- Figure 5 legend: line 604, change Nb67 by Nb61.5

Thanks for spotting this - mistake corrected

7- Figure 5C: neither P2-M4 nor M2, both mentioned in the legend (line 607-608), appear in the proposed figure.

Thanks for spotting this – the figure was changed in draft but not the legend- mistake now corrected

8- Figure 5D: this panel is supposed to show a view of interactions between N61 and the CAP side, but here the nanobody is labelled N57Nb. Figure 5D: this panel is supposed to represent a view of the interactions between N61 and the CAP side, but it shows a conformational change of M4 with and without the nanobody, which is not shown in this view. To be modified.

Now corrected – the figure was updated in draft, but not the legend.

9- Supplementary Figure S1: The figure is in negatif... (probably an editing problem) and therefore impossible to revise.

Not sure about this comment – the pdf seemed OK to us and other reviewers – we will double check that the uploaded version is compatible with different pdf viewers.

10- Supplementary Figure S2: The choice of colors to represent the different structures and electron densities is not very judicious and does not make it easy to distinguish nanobody and channel. The legend should explain the significance of the indices A, B, C and D assigned to the various residues shown.

The indices a-d etc refer to the chains in the pdf. Some additional labels have been added to attempt to clarify this – the main purpose of this figure is to illustrate the quality of the electron density rather than visualise interactions themselves.

11- Supplementary Figure s3 : Image quality is very poor. It is essential to improve the quality so that the reader can properly appreciate the relevance of the conclusion.

As stated above we have attempted to make sure this is converted to pdf at the highest resolution possible, but the principal results are negative results i.e., empty black boxes and so there is nothing to see in nearly all of the 'results' boxes.

Reviewer #2 (Remarks to the Author):

K channels are important drug targets for many diseases but obtaining high selectivity with small molecules can be challenging. This study from Rodstrom et al instead studies nanobodies as alternatives to small molecules, specifically against TREK-2 channels, to create advanced pharmacological tools that could be used both for basic discovery research and explored for therapeutic potential.

Rodstrom et al used purified TREK-2 protein injected into llamas to raise a panel of nanobody binders. From the panel four nanobodies of interest were explored in more detail, one neutral, one inhibitor and two activators. The study uses X-ray crystallography and cryo-EM to reveal the binding modes of each nanobody and provide insights into molecular mechanisms of action. This is supported by a thorough electrophysiological characterisation of function in combination with mutagenesis studies to confirm the roles of residues.

Overall, this study combines a variety of techniques to identify important new pharmacological tools for potassium channel modulation whilst explaining molecular mechanisms of action. It has broad conceptual relevance to the ion channels field for understanding antibody modulation and informing on the advancement of antibody modulators, making it suitable for publication in Nature Comms.

Minor comments are:

- Line 196: What kind of interaction? Polar? Between which parts? Side chains? Backbones?

These are polar interactions – now clarified in text.

- Line 201: What kind of interaction? Polar? Between which parts? Side chains? Backbones?

These are polar interactions – now clarified in text.

- The “Selective activatory nanobody” section. It would be nice to provide an additional figure pinpointing any residues involved in interactions that differ between TREK-1 and 2, to explain the molecular basis of selectivity. I do not suggest any need for mutagenesis studies to verify this, but inclusion of a figure would be insightful. If there is no clear molecular explanation then it would still be useful and interesting to discuss this.

The interactions between the side of the Cap domains, the loops and the top of M3 are extensive and difficult to visualise in a comprehensive way – we have now clarified this in the text and listed some of them – many are unique but these would be difficult to dissect in terms of ones which are structurally relevant due to the number involved. Instead, we selected some those we think may be functionally relevant and showed these - in particular the ones with the top of M3 which we also now discuss in more details (see response to Reviewer 1)

- The “inhibitory nanobody with dual effects” section. I did not understand why the word “dual” was present. So far as I can tell the nanobody is an inhibitor and does not have dual effects.

In this context ‘dual’ referred to the mechanism of action i.e. both direct block and allosteric - we have now clarified this in the text.

- The “A biparatropic inhibitory nanobody with improved efficacy” section. As I understand efficacy it refers to strength of maximal effect, e.g. does an inhibitory nanobody at saturating concentration only inhibit channels by 20% or 80 %, etc? Whereas what you are looking at is sensitivity of effect – concentration required to achieve 50 % inhibition. To avoid confusion in the field replace efficacy with sensitivity.

We agree this term is not strictly correct and have now clarified the section which refers to the higher affinity of this engineered nanobody for its target.

Reviewer #1 (Remarks to the Author):

Our comments and suggestions have been taken into account, and the authors' responses are well argued. By modifying the manuscript as suggested (particularly the discussion and bibliographical references), the authors have given the document the degree of relevance that the previous version lacked for immediate publication.

Reviewer #2 (Remarks to the Author):

The authors have fully addressed my comments.